# TransFusion: Contrastive Learning with Attention Layers

## Abstract

This paper proposes *TransFusion*, a novel framework for training attention-based neural networks to extract useful features for downstream classification tasks. *TransFusion* leverages the fusion-like behavior of the self-attention mechanism, where samples from the same cluster have higher attention scores and gradually converge. In the pursuit of deriving meaningful features, *TransFusion* adopts a strategy of training with affinity matrices, effectively capturing the resemblances among samples within the same class. In the context of classification-related tasks with limited understanding of the Attention layer's functionality, we offer theoretical insights into the actual behavior of each layer. Our main result demonstrates *TransFusion*'s effectiveness at fusing data points within the same cluster, while simultaneously ensuring careful management of noise levels. Furthermore, experimental results indicate that *TransFusion* successfully extracts features that isolate clusters from complex real-world data, leading to improved classification accuracy in downstream tasks.

## 1 Introduction

Contrastive Learning (CL) has recently garnered significant attention due to its effectiveness in training feature extraction models without the need for labeled data. Along this trajectory, several renowned models have been introduced, including SimCLRChen et al. (2020), Contrastive Multi-view Coding (CMC) Tian et al. (2020a), VICRegBardes et al. (2021), BarLowTwinsZbontar et al. (2021), and Wu et al. (2018); Henaff (2020). These approaches typically share a common framework: during training, the objective is to minimize the distance between augmented versions of images from the same source while simultaneously maximizing the distance between images from different sources. Following the training phase, the model is commonly combined with a feed-forward neural (FFN) decoder to fine-tune its performance using labeled data. Empirical evidence demonstrates that these models can achieve performance levels comparable to fully-supervised models, even when trained with a relatively limited amount of labels (approximately $10\%$) on moderate to large datasets Jaiswal et al. (2020). Amidst the recent surge in follow-up research, the research community has predominantly focused on two critical aspects: 1) determining the optimal augmentation strategy Wang & Qi (2022); Saunshi et al. (2022); Tian et al. (2020b); Xiao et al. (2020); Wang et al. (2022) and 2) identifying the most effective loss function Yeh et al. (2022); Yang et al. (2022a;b); Zhu et al. (2022); Cui et al. (2021).

To address these key questions, we introduce a novel framework *TransFusion* that allows a more analytic and predictable embedding learning process. Specifically, we architect a fusion model seamlessly harnessed by the self-attention mechanism, which assigns higher attention scores to samples belonging to the same cluster. After the self-attention operation, a weighted-sum operation further fuses samples from the same cluster, bringing them closer in the embedding space (see example in Table 1). Our research delves into the limitations inherent in the widely acclaimed loss function, *InfoNCE* Oord et al. (2018), and offers a symmetric yet computationally stable alternative founded on *Jensen-Shannon Divergence* Lin (1991). This new loss function not only mitigates the instability issues associated with *InfoNCE* but also accommodates the inclusion of more than two images from the same class within each batch. Furthermore, we establish a theoretical upper boundary on the augmentation required for achieving successful fusion in the learning process. This achievement is made possible by the unique and insightful architecture of *TransFusion*'s trainable fusion process, which enables a layer-wise fusion of embeddings, facilitating a more flexible and adaptable learning

| Input | Layer 1 | Layer 2 | Layer 3 | Layer 4 |
|---|---|---|---|---|

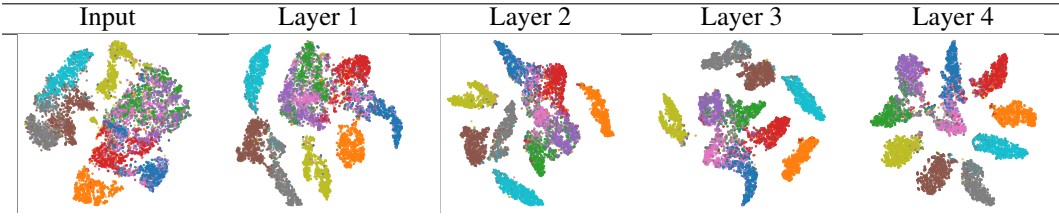

Table 1: Demonstration of *TransFusion*'s layer by layer fusion effect, leading to the emergence of notably denser and distinct clusters corresponding to individual classes. The embeddings are retrieved from a 5-layer *TransFusion* model on FashionMNIST, and plotted using T-SNE visualization.

journey. Compared to other CL models, whose theoretical aspects in the context of classification tasks remain largely unknown, our theoretical results show the insights that each layer excels at effectively fusing data points within the same cluster, while simultaneously ensuring careful management of noise levels.

To leverage the fusion-like behavior exhibited by the Attention layer, *TransFusion* utilizes a series of modified attention layers, where the traditional softmax function is replaced with an element-wise ReLU function. The model takes a group of samples as input and generates a matrix of attention scores as the final output. Subsequently, we provide the model with a target affinity matrix that indicates whether each pair of images belongs to the same or different class. By minimizing the *Jensen-Shannon Divergence* between the model's output and the target affinity matrix, each layer of the model progressively fuses the input to produce embeddings that are denser but distinct across different classes.

This design offers several noteworthy advantages, with its predictability standing out as the foremost benefit. To validate this assertion, we conducted an extensive theoretical analysis. Our findings conclusively reveal that, as long as the noise remains within a controlled threshold, each layer within the *TransFusion* model actively contributes to the fusion process. This insight can be readily translated to establish an upper limit on the required augmentation for achieving a successful fusion process when employing images as the direct input. Furthermore, it elucidates the influence of class structure (i.e., intra-cluster and inter-cluster distances) on the effectiveness of fusion. In contrast to other theories regarding Attention-based models, which suggest that specific layers require specific weights to fulfill specific tasks, all layers in the *TransFusion* model perform a simple and repetitive task: fusing points to achieve an improved embedding space. By executing these straightforward tasks that align with expectations, the *TransFusion* model exhibits enhanced stability and predictable behavior, making it well-suited for practical applications.

Furthermore, *TransFusion* introduces a heightened degree of adaptability in its management of the loss function. This is because the fusion process can be further conceptualized as a relaxation of the loss function, which assumes that all samples within the same class should exhibit precisely identical similarity. However, this approach disregards the inherent characteristics of datasets, wherein certain samples may exhibit varying degrees of proximity or distance from each other even within the same class. In contrast, *TransFusion* alleviates this stringent constraint by allowing samples to organize into sub-clusters across different layers, subsequently consolidating the centroids of these sub-clusters into a unified cluster. This augments the interpretive flexibility of the embeddings. Additionally, *TransFusion* is purposefully designed as a plug-in learning model, seamlessly capable of integration with any existing classification network.

**Paper Organization.** We provide a formal introduction to *TransFusion* and its key insights in Section 2. Subsequently, we conduct a comprehensive theoretical analysis of *TransFusion* in Section 3. Moving forward, we delve into the landscape of relevant research pertaining to neural network embedding learning in Section 4. To validate the efficacy of our approach, we present the experimental results in Section 5.

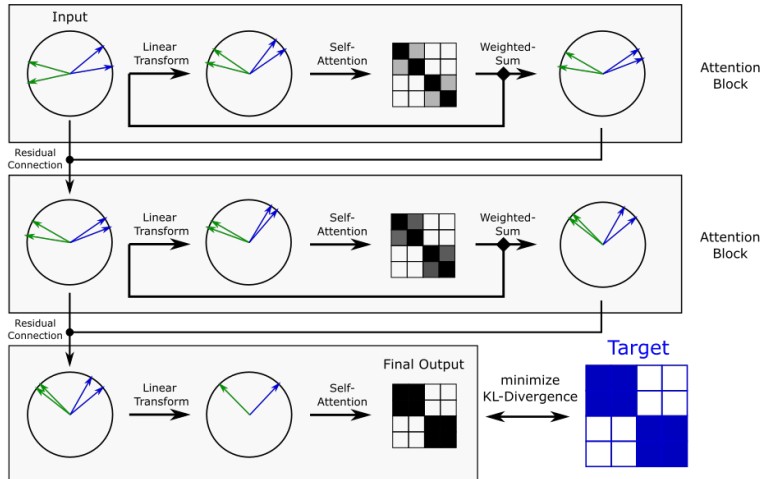

Figure 1: The architecture of the *TransFusion* Model. Its mid layers consist of regular attention blocks with ReLU replacing the softmax function for the affinity matrix. In the final layer, we train the model to minimize the KL-Divergence between predicted affinity matrix and the provided affinity matrix-.

## 2 TRANSFUSION

Let us consider the input data as a collection of samples $\mathbf{x} \in \mathbb{R}^m$ that are associated with various classes. The training set is represented by the input matrix $\mathbf{X} \in \mathbb{R}^{n \times m}$, where $m$ represents the ambient dimension, and $n$ represents the number of samples, with each row corresponding to a data sample:

$$\mathbf{X} = [\mathbf{x}_1, \ldots, \mathbf{x}_n]^\top$$

Our main goal is to create embeddings that effectively distinguish between different classes. This implies that samples belonging to the same class should be closer to each other in distance compared to samples from different classes. To achieve this objective, we introduce the *TransFusion* model (see Figure 1) with customized Attention blocks. The *TransFusion* model takes a set of input samples and generates a non-negative affinity matrix, which reflects the similarities between pairs of samples. Higher values in the affinity matrix at the $i, j$'th entry indicate stronger similarities between sample $i$ and sample $j$. To train the model, we create a target affinity matrix based on the ground-truth labels and then minimize the Kullback-Leibler Divergence (KL Divergence) between the output affinity matrix and the target affinity matrix. After identifying the best model that minimizes the divergence between the target and output affinity matrices, we extract the output of the second-to-last layer to obtain the final embeddings.

To gain deeper insight into the *TransFusion* model's functioning, let us delve into the details of the Attention Head, which comprises three key components: Query $\mathbf{Q}$, Key $\mathbf{K}$, and Value $\mathbf{V}$. Each of these components undergoes a linear transformation based on the input $\mathbf{X}$. The output of an attention head is obtained by applying a similarity-weighted sum to the Values $\mathbf{V}$:

$$Attn(\mathbf{X}) = \sigma(\mathbf{Q}\mathbf{K}^T)\mathbf{V} = \sigma((\mathbf{X}\mathbf{W}_Q)(\mathbf{X}\mathbf{W}_K)^\top)(\mathbf{X}\mathbf{W}_V)$$

where $\mathbf{W}_Q$, $\mathbf{W}_K$, and $\mathbf{W}_V \in \mathbb{R}^{m \times m}$ represent the learn-able parameters for the transformations, and $\sigma$ denotes the column-wise softmax function. Under this mechanism, if the model learns a effective linear transformation of Query $\mathbf{Q}$ and Key $\mathbf{K}$, the similarity between samples from the same class can be high, whereas the similarity from the different class can be low. Using this as the weight in the next weighted-sum calculation, samples from the same class shall be able to fuse closer to each other. However, the drawback of this model is that the softmax function $\sigma$ tends to over-sparsify the similarty, which makes limited samples fuse at a time, and thus less effective fusion into clusters.

To leverage this fusion effect, we change the softmax function $\sigma$ to elementalwise ReLU. This gives us a few advantages: 1) Increased clustering flexibility: By adopting elementalwise ReLU, multiple

elements with high similarity scores can fuse into the same cluster. This approach facilitates the integration of a broader range of elements into clusters. 2) Enhanced model non-linearity: It is important to note that the model does not necessarily require a feed-forward layer immediately after each attention block. However, by incorporating an activation function at this stage, we introduce non-linearity into the model, thereby fortifying its resilience and adaptability.

Formally, denote $\mathbf{X}^\ell$ as the input of the $\ell$'th block of the model in the *TransFusion*, then $\ell$'th block can be defined as

$$\mathbf{A}^\ell := (\mathbf{X}^\ell \mathbf{W}_Q^\ell)(\mathbf{X}^\ell \mathbf{W}_K^\ell)^\top \tag{1}$$

$$\mathbf{X}^{\ell+1} := \mathrm{ReLu}(\mathbf{A}^\ell)(\mathbf{X}^\ell \mathbf{W}_V^\ell) + \mathbf{X}^\ell \tag{2}$$

Denote $d$ is number of blocks intended for the model, then the final affinity matrix output is defined as $\mathbf{A}^d$, and the final embeddings are $\mathbf{X}^{d-1}$.

The final block of the model is what make *TransFusion* differ from any other attention-based networks. Our objective is to ensure that the affinity matrix $\mathbf{A}$ equals training target affinity matrix $\mathbf{Y}$, indicating successful capture of the cluster structure represented by $\mathbf{Y}$. For multi-class classification task, $[\mathbf{Y}]_{ij}$ can be a indicator of whether sample $i, j$ are in the same class. Moreover, for classes with hierarchical structures, $[\mathbf{Y}]_{ij}$ can also be modified to represent that. Specifically,

$$[\mathbf{Y}]_{ij} = \begin{cases} 1 & i, j \text{ in the same class and } i \neq j \\ \varepsilon^\tau & i, j \text{ in the same parent class but } \tau \text{ depths away} \\ 0 & \text{Otherwise} \end{cases} \tag{3}$$

where $\varepsilon$ is a decay constant. An example can be referred to Figure 2.

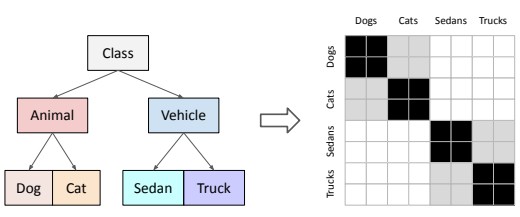

Figure 2: Contract Affinity matrix $\mathbf{Y}$ based on the hierarchical structure of the data. Black pixel in the affinity matrix represents 1, and gray pixel represents $\varepsilon$.

To attain our objective, one viable approach is to minimize the *Kullback-Leibler Divergence* Kullback & Leibler (1951) between $\mathbf{Y}$ and $g(\mathbf{A})$, where $g : \mathbb{R}^{n \times n} \to \mathbb{R}^{n \times n}$ represents a function that transforms affinities into probabilities. Remarkably, if we set $g(\mathbf{A})$ to be the row-wise *Softmax* $\sigma$, our loss with a one-layer *TransFusion* model behaves identically to *InfoNCE*. The detailed proof is available in Appendix A.

However, there is a significant concern with the use of *KL-Divergence* due to its inherent asymmetry. This issue becomes particularly problematic in this context, as a majority of entries in $\mathbf{Y}$ are filled with zeros. To see this, take any pair of samples $i$ and $j$ belonging to different classes, the loss term $\left([\mathbf{Y}]_{i,j} \log \frac{[\mathbf{Y}]_{i,j}}{[g(\mathbf{A})]_{i,j}}\right)$ prevents the model from penalizing the positive value of the affinity term $[g(\mathbf{A})]_{i,j}$ as the target affinity is zero, i.e. $[\mathbf{Y}]_{i,j} = 0$. Although the normalization within $g$ might indirectly influence $[g(\mathbf{A})]_{i,j}$ to be small, this effect is relatively minor compared to the influence other terms. In simpler terms, this loss function strongly encourages embeddings from the same class to be close to each other while largely disregarding the distances between embeddings from different classes.

To solve this issue, we use *Jensen-Shannon Divergence* , which is a symmetric loss that avoids singularity with entries being zero. Specifically,

$$\mathcal{L}_{\mathrm{TF}} := \mathcal{D}(\mathbf{Y} || g(\mathbf{A}) + \mathbf{Y}) + \mathcal{D}(g(\mathbf{A}) || g(\mathbf{A}) + \mathbf{Y}) \tag{4}$$

where $\mathcal{D}$ denotes *KL-Divergence* loss. As for the selection of $g(\mathbf{A})$, we deliberately opt for using row-normalized of $\mathbf{A}^2$, i.e.

$$g(\mathbf{A}) := \mathrm{Row\text{-}Normalize}(\mathbf{A}^2) \tag{5}$$

The rationale behind this choice lies in the potential existence of negative affinity scores within $\mathbf{A}$, a scenario where traditional *Softmax* treatment might designate negative values as indicative of "low

probability." However, in our unique context, such negativity does not necessarily signify insignificance. To illustrate, consider the case where Query $\mathbf{q}_i$ perfectly captures the subspace of Key $\mathbf{k}_j$. We can ensure that the magnitude of the projection from $\mathbf{q}_i$ to $\mathbf{k}_j$ is large, possibly equal to $||\mathbf{q}_i||_2$. However, we cannot tell whether two vectors face the same direction or exact opposite directions. This latter scenario results in a negative projection value. Consequently, the act of converting negative values to low probabilities inadvertently constricts the model's capacity to flexibly capture the intrinsic substructure of the Keys.

Through our empirical analysis, we have uncovered a noteworthy enhancement in stability by adopting our custom loss function (5) in comparison to alternative choices, including *InfoNCE*. This improvement is evident in both performance and resilience to changes in the learning rate. For a more detailed exploration and comprehensive insights, please refer to Appendix B.

## 3 THEORETICAL GUARANTEES

Recall that the input matrix $\mathbf{X}$ has dimensions $n \times m$, where $m$ represents the ambient dimension and $n$ represents the number of samples. It is common for datasets with multiple classes to exhibit some form of clustering. In our study, we specifically focus on subspace clusters, where samples belonging to the same class can be linearly dependent on each other.

Specifically, we assume that the data matrix $\mathbf{X}$ can be decomposed into a collection of low-dimensional subspaces $\{\mathcal{U}_1, ..., \mathcal{U}_\kappa\}$, where $\kappa \ll n$, and each row of $\mathbf{X}$ lies in one of these subspaces. We assume that the rank of each subspace is at most $r$. It is easy to see that for each subspace $\mathcal{U}_i$, there exists a unit vector $\mathbf{u}_i^\perp$ that is orthogonal to $\mathcal{U}_i$, such that for any $\mathbf{x}_j$ not belonging to the subspace $\mathcal{U}_i$, the inner product $\mathbf{x}_j^\top \mathbf{u}_i^\perp$ is greater than or equal to a certain threshold $\rho$, i.e.,

$$\forall \mathbf{x}_j \notin \text{Span}(\mathcal{U}_i), \quad \mathbf{x}_j^\top \mathbf{u}_i^\perp \geq \rho. \tag{6}$$

In the upcoming section, we will provide a theoretical proof that assures the existence of solutions for the *TransFusion* model, enabling it to perform fusion not only on noise-free data but also on data with controlled noise.

### 3.1 NOISE-FREE FUSION

Recall that the last layer of the *TransFusion* produces the predicted affinity matrix by:

$$\mathbf{A} = (\mathbf{X}\mathbf{W}_Q)(\mathbf{X}\mathbf{W}_K)^\top = \begin{bmatrix} \mathbf{x}_1^\top \mathbf{W}_Q \\ ... \\ \mathbf{x}_n^\top \mathbf{W}_Q \end{bmatrix} \begin{bmatrix} \mathbf{W}_K^\top \mathbf{x}_1 & ... & \mathbf{W}_K^\top \mathbf{x}_n \end{bmatrix}.$$

where $\mathbf{W}^\top \mathbf{x}_i \in \mathbb{R}^m$ can be understood as a linear transformation of $\mathbf{x}_i$. Picking any entry from $\mathbf{A}$, we can see that $[\mathbf{A}]_{i,j}$ measures the cosine similarity of $\mathbf{x}_i$ and $\mathbf{x}_j$ after linear transformation, i.e.

$$[\mathbf{A}]_{i,j} = (\mathbf{W}_Q^\top \mathbf{x}_i)^\top (\mathbf{W}_K^\top \mathbf{x}_j).$$

Our first theoretical result, summarized in the following theorem, shows that if each sample in $\mathbf{X}$ comes from one of the subspaces in the union $\{\mathcal{U}_1, ..., \mathcal{U}_\kappa\}$, we can always find a set of $(\mathbf{W}_Q^*, \mathbf{W}_K^*)$ that enables the similarity score matrix $\mathbf{A}$ to separate clusters.

---

**Theorem 1.** *Given a collection of input samples* $\mathbf{X}$*, where every sample* $\mathbf{x}_i$ *comes from one subspaces among* $\{\mathcal{U}_1, ..., \mathcal{U}_\kappa\}$*, there always exists a pair of parameters* $(\mathbf{W}_Q^*, \mathbf{W}_K^*)$ *such that the affinity matrix* $\mathbf{A}$ *calculated by (1) has a block-diagonal form, ie.*

$$\begin{cases} [\mathbf{A}]_{i,j} = 0, & \mathbf{x}_i, \mathbf{x}_j \text{ in different clusters} \\ [\mathbf{A}]_{i,j} > 2\rho^2, & \mathbf{x}_i, \mathbf{x}_j \text{ in the same cluster} \end{cases}$$

*where* $\rho$ *is the lowerbound of cross-class projection defined in* (6)

---

The proof of Theorem 1 proceeds by selecting each column within $\mathbf{W}_Q$ and $\mathbf{W}_K$ to be orthogonal to one of the subspaces from the set $\{\mathcal{U}_1, \dots, \mathcal{U}_\kappa\}$. Subsequently, the proof establishes a lower bound on the projection of samples onto the columns of $\mathbf{W}_Q$ and $\mathbf{W}_K$ using the expression given in (6). For a comprehensive and detailed proof, please refer to the Appendix in Section C.

### 3.2 NOISY FUSION

To address the presence of noise, the clear block-diagonal structure of $\mathbf{A}$ may not be immediately apparent within a single layer. Therefore, our objective is to progressively enhance the block-diagonal nature of $\mathbf{A}$ with each subsequent layer. By stacking a sufficient number of layers, the final layer's output $\mathbf{A}$ should possess the desired property to effectively recognizing clusters. To achieve this, we introduce a standard measurement that quantifies the degree of similarity between $\mathbf{A}$ and a perfect block-diagonal matrix.

**Definition 1.** *The* sharpness *of the similarity score matrix* $\mathbf{A}$ *is defined as the infimum of the ratio between the similarity scores of two points within the same cluster and the similarity scores of two points belonging to different clusters, i.e.,*

$$\mathcal{S}(\mathbf{A}) := \inf_{i,j,k,h} \frac{[\mathbf{A}]_{i,j}}{[\mathbf{A}]_{k,h}},$$

*where* $\mathbf{x}_i, \mathbf{x}_j$ *are from the same cluster and* $\mathbf{x}_k, \mathbf{x}_h$ *are from different clusters.*

Here, denote the noisy version of $\mathbf{X}$ as $\tilde{\mathbf{X}}$. Each sample of $\tilde{\mathbf{X}}$ has a non-negative cosine similarity with the corresponding sample in $\mathbf{X}$. The cosine similarity is lower-bounded by a universal constant $\varepsilon \in [0, 1]$:

$$\mathbf{x}_i^\top \tilde{\mathbf{x}}_i \geq (1 - \varepsilon)$$

where $\mathbf{x}_i, \tilde{\mathbf{x}}_i$ are all normalized unit-length vector. With this setting, we can further lower-bound the projection of $\tilde{\mathbf{x}}_j$ onto the orthogonal complement $\mathbf{u}_i^\perp$:

1) When $\mathbf{x}_j$ is in the span of $\mathbf{U}_i$,

$$\tilde{\mathbf{x}}_j^\top \mathbf{u}_i^\perp \leq \sqrt{(1 - (1-\varepsilon)^2)} =: \delta;$$

2) When $\mathbf{x}_j$ is not in the span of $\mathbf{U}_i$,

$$\tilde{\mathbf{x}}_j^\top \mathbf{u}_i^\perp \geq (1-\varepsilon)\rho - \sqrt{(1-(1-\varepsilon)^2)(1-\rho^2)} =: \Delta.$$

Intuitively, $\delta$ represents the maximum cosine similarity observed between input samples and the complementary set of the same class, and $\Delta$ represents the minimum cosine similarity observed between input samples and the complementary set of a different class.

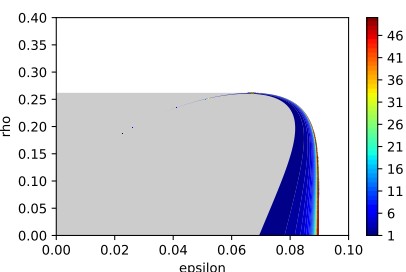

Figure 3: Validation of Theorem 2 through Numerical Analysis: The highlighted region depicts where the sharpness increment $\gamma \geq 1$. The colorbar quantifies the exact increment in sharpness. The shaded area further signifies the fusion area's effectiveness, where the $(\rho, \epsilon)$ pair adheres to more lenient conditions.

Building upon this foundation, we can extend our control over the advancement of the fusion process within each block. In particular, we substantiate that, mirroring the configuration of noise-free data, every *TransFusion* block possesses the capacity to enhance the **sharpness** of its affinity matrix compared to the preceding layer. This enhancement is quantifiable by a constant factor determined by the noise-level constant $\varepsilon$ and the lower bound of projection denoted by $\rho$.

**Theorem 2.** *Given a collection of input samples* $\tilde{\mathbf{X}}$*, where each sample* $\tilde{\mathbf{x}}_i$ *lies near one of the subspaces among* $\{\mathcal{U}_1, ..., \mathcal{U}_\kappa\}$*, there always exists a pair of parameter* $(\mathbf{W}_Q^*, \mathbf{W}_K^*)$ *such that each attention block increases the* **sharpness** *of its similarity matrix* $\mathbf{A}$ *calculated by* (1) *by at least a factor of* $\gamma := \frac{(\alpha^2+\beta^2)\Delta^2\beta}{2\alpha(\alpha^2+\beta^2)\delta + 2\alpha^2\beta(1+\delta^2)}$*, where* $\alpha := 2(\Delta^2 + \delta^2)$ *and* $\beta := 4\Delta\delta$*.*

In words, Theorem 2 conveys that when the within-cluster distance $\delta$ is sufficiently small and the between-cluster distance $\Delta$ is suitably large, there will be a notable increase in sharpness by a factor greater than 1. This will ultimately lead to an enhanced block-diagonal pattern within the similarity matrix $\mathbf{A}$. Figure 3 shows the sharpness increment $\gamma$ as a function of $\rho$ and $\epsilon$. These results show that for specific ranges of $(\varepsilon, \rho)$, the bound exceeds 1, thereby ensuring fusion, as claimed by the Theorem.

The proof of Theorem 2 follows by the same arguments as that of Theorem 1, except the off-diagonal blocks are upper bounded by $\delta$ instead of zero, and the diagonal blocks are lower bounded by $\Delta$. The detailed proof can be found in the Appendix D.

## 4 RELATED WORK

Numerous endeavors have been made to extract valuable embeddings from images using contrastive learning methods. These efforts can generally be classified into two primary categories: self-supervised and supervised learning. In the following segment, we will delve into pertinent research within each of these categories.

### 4.1 SELF-SUPERVISED CONTRAST LEARNING

In the domain of contrastive learning, each individual image is treated as a distinct class, and the model is honed to discern augmented versions of the same image amidst a backdrop of other images. Approaches conceived within this category conventionally revolve around designating an anchor image and then generating augmentations of this anchor image. In tandem with the other images, the model is instructed to intensify the similarity between the anchor image and its augmentations, while concurrently diminishing the similarity between the anchor image and the remaining images. A noteworthy milestone emerges in the form of a seminal study conducted by Wu et al. (2018). This work introduces a non-parametric approach for gauging the similarity between features, thereby elevating accuracy levels across datasets like CIFAR10 and ImageNet. Another significant contribution comes from Hjelm et al. (2018), who showcase the potential of maximizing mutual information between an input and the output of a neural network encoder. Similarly, Tian et al. (2020a) build upon a comparable method to amplify the mutual information across diverse channels of a scene's view. The results of their experimentation corroborate the efficacy of this approach. Intriguingly, the utilization of contrastive learning has ushered in a remarkable shift in training larger networks with significantly fewer labeled data points, all the while achieving competitive outcomes on prominent datasets such as Imagenet Chen et al. (2020) and PASCAL VOC Henaff (2020). A more recent contribution that stands out is the work undertaken by Balestriero & LeCun (2022). Remarkably, this study unveils the closed-form optimal representation and network parameters within the linear regime for prevalent self-supervised learning approaches, including VICReg Bardes et al. (2021), SimCLR Chen et al. (2020), and BarlowTwins Zbontar et al. (2021).

### 4.2 SUPERVISED CONTRAST LEARNING

Contrastive Learning finds its application not only within unsupervised settings but also extends its influence into the realm of supervised learning. Rather than treating each image as an isolated class, this approach harnesses the power of labels. Here, the model takes advantage of these labels by encouraging comparisons between images spanning various classes. The goal is to minimize the distance between embeddings belonging to the same class. One notable contribution in this direction, presented by Khosla et al. (2020), demonstrates that by pairing images from the same class as the anchor image, the model exhibits enhanced performance across datasets such as CIFAR10 and CIFAR100. Additionally, it showcases increased robustness when faced with images corrupted by various factors. Zheng et al. (2021) propose an innovative methodology encompassing a dual-pronged approach. They employ a network to grasp the nuances of the discriminative task, complemented by a graph-based model that effectively brings together similar samples. This combined setup empowers the model to achieve comparable performance levels while utilizing as little as $10\%$ of the available labels. Addressing the challenge of imbalanced class label distributions, Cui et al. (2021) introduce a novel model called Parametric Contrastive Learning (PaCo). This model not only tackles the issue but also establishes itself as a leader in the domain of long-tailed recognition.

In the domain of few-shot embedding model training, Liu et al. (2021) delve into the combination of contrastive learning with Noise Contrastive Estimation (NCE) within a supervised framework. This strategic integration leads to commendable performance on the miniImageNet dataset.

# 5 EXPERIMENT

In this section, we demonstrate the practical utility of *TransFusion* through experiments conducted on real-world datasets. Our focus is to showcase the effectiveness of *TransFusion* in various scenarios. First, we explore *TransFusion*'s layer-by-layer fusion impact using two datasets, namely FashionMNIST, and CIFAR-10. FashionMNIST comprise 10 categories of black and white and grayscale images respectively, each sized at 28 x 28. Meanwhile, CIFAR-10 consists of color images with dimensions of 32 x 32, encompassing 10 classes. Subsequently, we engage in classification tasks using embeddings derived from ResNet18 when trained with *TransFusion* and alternative techniques.

## 5.1 VISUALIZING *TransFusion*'S FUSION PROCESS

*TransFusion* operates on the fundamental concept of utilizing attention mechanisms to modify the embedding space, resulting in closer distances between related objects and farther distances between non-related objects. Through this fusion process, clusters become more concentrated and isolated, leading to enhanced distinctiveness among individual clusters. As demonstrated in the previous section, our theoretical framework establishes the result that each layer in the model should consistently amplify the overall cluster structure. In this section, we substantiate this claim through empirical experiments conducted on real-world datasets.

To provide a visual representation of the fusion's influence, we build a 5-layer *TransFusion* model for the fashionMNIST datasets, and 15-layer model for CIFAR10. For training, we utilize Adam optimizers and conduct 200 epochs. Following the training phase, we evaluate the models using unseen test datasets. During the evaluation, we extract intermediate outputs from each layer and generate embeddings through t-Distributed Stochastic Neighbor Embedding (t-SNE) Van der Maaten & Hinton (2008), which minimizes the discrepancy between similarity matrices to transform high-dimensional data into a lower-dimensional space while preserving the relationships and structure within the data. Moreover, we quantitatively evaluate the effectiveness of the fusion process by calculating the Silhouette Index (SI) for each layer Rousseeuw (1987). The Silhouette Index (between $-1$ and 1) measures the quality of clustering, considering both the cohesion within clusters and the separation between different clusters. A higher Silhouette Index indicates well-defined and separated clusters. The results obtained from this process are presented in Table 1 and 2.

| Input | Layer 5 | Layer 10 | Layer 12 | Layer 14 |
|-------|---------|----------|----------|----------|
| -0.14 | -0.08 | -0.05 | -0.03 | -0.02 |

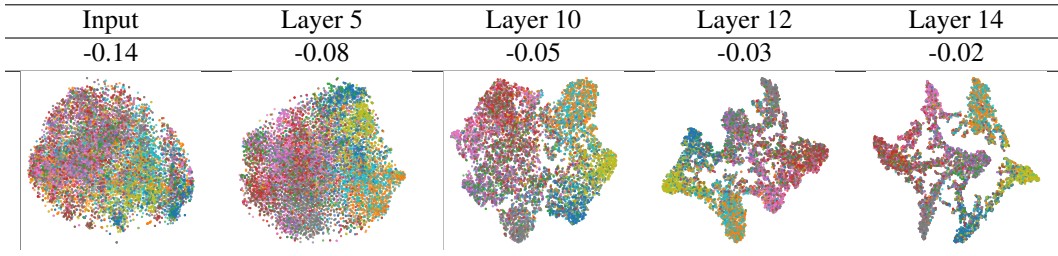

Table 2: T-SNE Embeddings with Silhouette Index of different *TransFusion* layer's output on CIFAR10 test dataset.

On the left-hand side of each graph, we observe the initial embeddings derived directly from the data. As we move from left to right within each row, we showcase embeddings from layers that are deeper into the models. In these graphs, each distinct class is represented by a different color, and the color scheme remains consistent throughout the respective datasets. Upon examining the graphs, two noticeable effects become apparent as the depth increases: i) The clusters exhibit greater separation, leading to an increased similarity ratio between samples belonging to the same class versus different classes. ii) The clusters become more condensed and exhibit reduced noise at the boundary, resulting in an enhancement of the infimum of the similarity ratio. These observations

| Model | Batch Size | SVM | K-NN (k=50) | K-NN (k=200) | Time (m) |
|---|---|---|---|---|---|
| SupCon | 128 | 0.8049 | 0.7907 | 0.7717 | 352 |
| SupCon | 1024 | 0.9056 | 0.9002 | 0.8931 | 641 |
| **TransFusion (this paper)** | 128 | **0.9309** | **0.9355** | **0.9350** | 368 |

Table 3: Accuracy Classification Results for CIFAR-10 Using *SupCon* and *TransFusion*. Each model was trained with ResNet18 as the underlying architecture and underwent a 200-epoch training process.

align with the theoretical understanding of *TransFusion*, which posits that each layer plays a role in sharpening the affinity matrix.

## 5.2 EMPIRICAL RESULTS ON CIFAR10

In this section, we embark on an empirical comparison between our approach and the Supervised Contrastive Learning method referred to as *SupCon* Khosla et al. (2020). *SupCon* essentially represents a modified variant of *InfoNCE*, but with the incorporation of supervised labels. This implies that, within each batch, images belonging to the same class are expected to exhibit high similarity, while those from different classes should display minimal similarity. Given that *SupCon* operates directly on the backbone of a classification network, we opt for an 18-layer ResNet He et al. (2016) as the backbone, which has been pre-trained on the ImageNet1K dataset. Subsequently, we train two separate models: one employing the *SupCon* method, and the other employing a one-layer *TransFusion* approach. Following this, we extract the intermediate outputs from the ResNet, which serve as embeddings. These embeddings are then employed to train a Support Vector Machine Hearst et al. (1998) using the training dataset, followed by conducting classification on the embeddings obtained from the test dataset. This procedure adheres to the standard practice for assessing the quality of fixed embeddings, in alignment with the guidance provided by Li et al. (2020). In addition to the SVM classification, we also perform K-Nearest-Neighbor (K-NN) classification with two different settings: one with $k = 50$ and the other with $k = 200$. This is executed using the same protocol as the SVM classification, as recommended by Yeh et al. (2022). For this experiment, we did not introduce any augmentation techniques. To enhance training performance, the input images were resized from dimensions of $(32, 32, 3)$ to $(224, 224, 3)$.

The results in Table 3 clearly indicate that *TransFusion* consistently outperforms *SupCon*, even when *SupCon* is trained with a larger batch size—a practice commonly thought to be more effective for contrastive learning, as noted in Chen et al. (2020). Remarkably, *ResNet18* achieves an accuracy of 0.935 when trained with full labels using cross-entropy loss in our own experiment. This underscores that *TransFusion* achieves comparable performance to a fully-supervised model with considerably less information, as it only necessitates pairwise relationships within each mini-batch for training.

## 6 CONCLUSION & FUTURE DIRECTION

In this paper, we introduced *TransFusion*, a novel framework for training attention-based neural networks that aims to extract informative features for downstream classification tasks. Our theoretical analysis demonstrates that each layer in the *TransFusion* model contributes to the fusion process, enhancing the embedding space's density and distinctiveness across different classes. Through experiments, we validated the efficacy of *TransFusion* by showcasing its ability to isolate clusters from complex real-world data.

This paper serves as the inaugural introduction to our proposed framework, laying the essential theoretical groundwork. While our current experimental results do not necessarily surpass the state-of-the-art benchmarks, they set the stage for our forthcoming research direction aimed at achieving this goal. Looking ahead, we envision promising avenues for future investigations, including the expansion of *TransFusion*'s application across a wider range of intricate and diverse datasets. Additionally, we plan to delve into refining and extending the model by incorporating image augmentation, experimenting with a blend of Convolutional Layers, and exploring methods for effective dimensional reduction.

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

## A APPENDIX: RELATION TO INFONCE

In this section, we demonstrate its relationship with well-known InfoNCE in both self-supervised and supervised manner. Specifically, for self-supervised learning, InfoNCE can be found in the form of:

$$\mathcal{L}_{\text{self}} = -\sum_{i \in I} \log \frac{\exp(\mathbf{z}_i \cdot \mathbf{z}_{j(i)}/\tau)}{\sum_{i \neq a} \exp(\mathbf{z}_i \cdot \mathbf{z}_b/\tau)} \tag{7}$$

where $I$ denote the collection of samples from the current training batch; $\tau$ is a temperature constant; $\mathbf{z}_i$ is the feature extraced from i'th sample; $j(i)$ corresponding to the positive example of sample $i$, which is usually generated by augmentation of sample $i$.

Since it's self-supervised learning, we know that there only exit 1 positive sample for each batch, which means that $|P(i)| = 1$, and $[\mathbf{Y}]_{ij}$ can only be either 0 or 1. This means that the loss can be simplified to:

$$\mathcal{L}_{\text{TF}} := \mathcal{D}(\mathbf{Y}||\sigma(\mathbf{A}^d/\tau)) = \sum_{i,j}[\mathbf{Y}]_{i,j} \log \frac{[\mathbf{Y}]_{i,j}}{[\sigma(\mathbf{A}^d/\tau)]_{i,j}} = \sum_{i \in I, j \in P(i)} -\log\left[\sigma(\mathbf{A}^d/\tau)\right]_{i,j}.$$

If we force the weight of Key and Query in the last layer to be the same $\mathbf{W}_Q^d = \mathbf{W}_K^d = \tilde{\mathbf{W}}$, then we can have embeddings $\mathbf{Z}$ denote as: $\mathbf{Z} := \mathbf{X}^\ell \tilde{\mathbf{W}}$, and

$$[\sigma(\mathbf{A}^d/\tau)]_{i,j} = \frac{\exp(\mathbf{z}_i \cdot \mathbf{z}_j/\tau)}{\sum_{i \neq a} \exp(\mathbf{z}_i \cdot \mathbf{z}_b/\tau)}$$

which is exactly the same as $\mathcal{L}_{\text{self}}$.

For supervised learning manner, referring to Khosla et al. (2020), the loss can be defined as

$$\mathcal{L}_{\text{sup}} = \sum_{i \in I} \frac{-1}{|P(i)|} \sum_{p \in P(i)} \log \frac{\exp(\mathbf{z}_i \cdot \mathbf{z}_p/\tau)}{\sum_{a \neq i} \exp(\mathbf{z}_i \cdot \mathbf{z}_b/\tau)} \tag{8}$$

where $P(i)$ denote the collection of samples from the same class as $i$'th sample. By pluging in (3) to (4), it's trivial that $\mathcal{L}_{\text{sup}} = \mathcal{L}_{\text{TF}}$.

## B APPENDIX: EMPIRICAL COMPARISON BETWEEN DIFFERENT LOSS FUNCTIONS

In this section, we undertake an empirical comparison among different loss functions. To conduct this comparison, we trained a one-layer Transfusion model on top of a pretrained 18 Layer ResNet He et al. (2016), employing each of the loss functions separately. Subsequently, we extracted the intermediate output from the ResNet as embeddings. These embeddings were then utilized to train a Support Vector Machine Hearst et al. (1998) on the training dataset, followed by performing classification on the embeddings from the test dataset. This procedure is a standard practice for evaluating the quality of the fixed embeddings, aligning with the recommendation provided by Li et al. (2020).

The Transfusion's training process follows a supervised contrastive learning approach, where the target affinity matrix is constructed based on the class labels of the samples. For this experiment, we did not introduce any augmentation techniques. To enhance training performance, the input images were resized from dimensions of $(32, 32, 3)$ to $(224, 224, 3)$.

Recall that one layer Transfusion model is essentially generating the cosine similarity between samples:

$$\mathbf{A} := (\mathbf{Z}\mathbf{W}_Q)(\mathbf{Z}\mathbf{W}_K)^\top \tag{9}$$

where $\mathbf{Z} \in \mathbb{R}^{n \times \tilde{m}}$ denotes the embeddings from the upstream ResNet18 model; $n$ denotes the number of samples; $\tilde{m}$ denotes the ambient dimension; $\mathbf{W}_Q, \mathbf{W}_K \in \mathbb{R}^{\tilde{m} \times \tilde{m}}$ denotes learn-able parameters. The loss functions we are interested are:

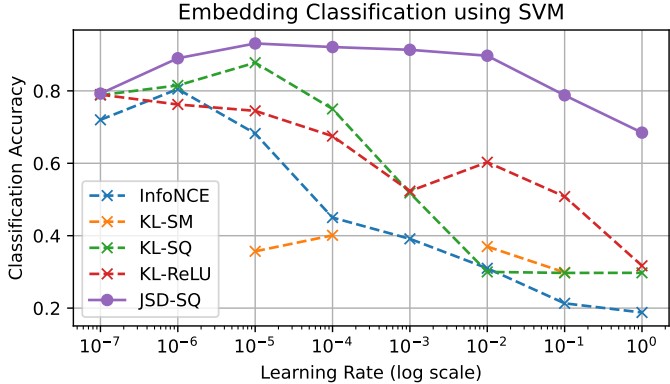

Figure 4: Precision in the encoding derived from a single-layer Transfusion model on top of ResNet18 He et al. (2016). This experiment was conducted on the CIFAR10 dataset, with ResNet18 pretrained on ImageNet1K as the underlying model.

- Supervised InfoNCE (8).
- KL-Divergence with Softmax (KL-SM): $\mathcal{L}_{\text{KL-SM}} := \mathcal{D}(\mathbf{Y}||\sigma(\mathbf{A}))$
- KL-Divergence with ReLU (KL-ReLU): $\mathcal{L}_{\text{KL-ReLU}} := \mathcal{D}(\mathbf{Y}||\text{Normalize}(\mathbf{A}_+))$
- KL-Divergence with Squared Affinity (KL-SQ): $\mathcal{L}_{\text{KL-SQ}} := \mathcal{D}(\mathbf{Y}||\text{Normalize}(\mathbf{A}^2))$
- Jensen-Shannon Divergence with Squared Affinity (JSD-SQ):

$$\mathcal{L}_{\text{JSD-SQ}} := \mathcal{D}(\mathbf{Y}||\text{Normalize}(\mathbf{A}^2) + \mathbf{Y}) + \mathcal{D}(\text{Normalized}(\mathbf{A}^2)||\text{Normalize}(\mathbf{A}^2) + \mathbf{Y})$$

The outcomes for each of the loss functions are depicted in Figure 4. From the results, it is evident that our custom loss function exhibits remarkable robustness across a broad spectrum of learning rates. It consistently maintains stable regions where classification accuracy remains consistently high. The next best performer is the KL-SQ, which shares similarities with the JSD-SQ. Notably, there are slight improvements when opting for JSD over KL. In contrast, it becomes apparent that the KL-SM function displays a heightened sensitivity to variations in the learning rate. Across numerous trials, we encountered challenges such as overflows, even after normalizing the values prior to the Softmax operation.

## C  APPENDIX: PROOF OF THEOREM 1

In words, Theorem 1 shows that the linear transformation of $\mathbf{x}_i$ and $\mathbf{x}_j$ has a cosine similarity of zero if they are from different clusters, and a positive similarity if they belong to the same cluster. To illustrate this, we present a simple example. Let $\mathbf{X}$ be a matrix with $n = 4$ samples and an ambient dimension of $m = 4$:

$$\mathbf{X} = \begin{bmatrix} \mathbf{x}_1^\top & \mathbf{x}_2^\top & \mathbf{x}_3^\top & \mathbf{x}_4^\top \end{bmatrix}^\top \in \mathbb{R}^{4\times 4}.$$

Consider two rank-1 subspaces, $\{\mathcal{U}_1, \mathcal{U}_2\}$, where $\mathbf{x}_1, \mathbf{x}_2 \in \mathcal{U}_1$, and $\mathbf{x}_3, \mathbf{x}_4 \in \mathcal{U}_2$. The bases of these subspaces can be represented as $\{\mathbf{u}_1, \mathbf{u}_2\} \in \mathbb{R}^4$. Define:

$$\mathbf{W}^* = \begin{bmatrix} \mathbf{u}_2^\perp & \mathbf{u}_2^\perp & \mathbf{u}_1^\perp & \mathbf{u}_1^\perp \end{bmatrix},$$

where $\mathbf{u}_1^\perp$ and $\mathbf{u}_2^\perp$ are bases orthogonal to $\mathcal{U}_1$ and $\mathcal{U}_2$, respectively, such that:

$$\mathbf{u}_1^\top \mathbf{u}_1^\perp = 0, \quad \mathbf{u}_2^\top \mathbf{u}_2^\perp = 0.$$

Using $\mathbf{W}^*$, we have:

$$\mathbf{X}\mathbf{W}^* = \begin{bmatrix} \mathbf{x}_1^\top \\ \mathbf{x}_2^\top \\ \mathbf{x}_3^\top \\ \mathbf{x}_4^\top \end{bmatrix} \begin{bmatrix} \mathbf{u}_2^\perp & \mathbf{u}_2^\perp & \mathbf{u}_1^\perp & \mathbf{u}_1^\perp \end{bmatrix} = \begin{bmatrix} \mathbf{x}_1^\top \mathbf{u}_2^\perp & \mathbf{x}_1^\top \mathbf{u}_2^\perp & 0 & 0 \\ \mathbf{x}_2^\top \mathbf{u}_2^\perp & \mathbf{x}_2^\top \mathbf{u}_2^\perp & 0 & 0 \\ 0 & 0 & \mathbf{x}_3^\top \mathbf{u}_1^\perp & \mathbf{x}_3^\top \mathbf{u}_1^\perp \\ 0 & 0 & \mathbf{x}_4^\top \mathbf{u}_1^\perp & \mathbf{x}_4^\top \mathbf{u}_1^\perp \end{bmatrix}$$

By setting $\mathbf{W}_K = \mathbf{W}_Q = \mathbf{W}^*$, we obtain:

$$\mathbf{A} = 2 \begin{bmatrix} (\mathbf{x}_1^\top \mathbf{u}_2^\perp)^2 & (\mathbf{x}_1^\top \mathbf{u}_2^\perp)^2 & 0 & 0 \\ (\mathbf{x}_2^\top \mathbf{u}_2^\perp)^2 & (\mathbf{x}_2^\top \mathbf{u}_2^\perp)^2 & 0 & 0 \\ 0 & 0 & (\mathbf{x}_3^\top \mathbf{u}_1^\perp)^2 & (\mathbf{x}_3^\top \mathbf{u}_1^\perp)^2 \\ 0 & 0 & (\mathbf{x}_4^\top \mathbf{u}_1^\perp)^2 & (\mathbf{x}_4^\top \mathbf{u}_1^\perp)^2 \end{bmatrix}$$

where each non-zero entry in $\mathbf{A}$ is lower-bounded by $2\rho^2$. Thus, the resulting matrix $\mathbf{A}$ effectively separates the two subspaces within $\mathbf{X}$.

The proof of Theorem 1 follows without complications by generalizing this example to larger bases, more samples, and more clusters, and is omitted here.

# D    APPENDIX: PROOF OF THEOREM 2

*Proof of Theorem 3.4.* Our proof of Theorem 2 will apply to our construction without the residual connection. The proof with residual connection follows the same logistic. Recall that

- The cosine similarity between noisy and original sample is lower-bounded by a universal constant $\varepsilon \in [0, 1]$:

$$\mathbf{x}_i^\top \tilde{\mathbf{x}}_i \geq (1 - \varepsilon)$$

- When $x_j$ is in the span of $\mathcal{U}_i$

$$\tilde{\mathbf{x}}_j^\top \mathbf{u}_i^\perp \leq \delta, \qquad \delta := \sqrt{(1 - (1 - \varepsilon)^2)}$$

- When $x_j$ is not in the span of $\mathcal{U}_i$,

$$\tilde{\mathbf{x}}_j^\top \mathbf{u}_i^\perp \geq \Delta, \qquad \Delta := (1 - \varepsilon)\rho - \sqrt{(1 - (1 - \varepsilon)^2)(1 - \rho^2)}$$

Let's go back the example, where the input has $n = 4$ samples with ambient dimension $m = 4$. Denote $\tilde{\mathbf{X}}^t$ as the $t$'th layer's input,

$$\tilde{\mathbf{X}}^t = \begin{bmatrix} \tilde{\mathbf{x}}_1^\top \\ \tilde{\mathbf{x}}_2^\top \\ \tilde{\mathbf{x}}_3^\top \\ \tilde{\mathbf{x}}_4^\top \end{bmatrix} \in \mathbb{R}^{4 \times 4}$$

Let there be 2 rank-1 subspace $\{\mathcal{U}_1, \mathcal{U}_2\}$, where $x_1, x_2 \in \mathcal{U}_1$, and $x_3, x_4 \in \mathcal{U}_2$. The basis of the subspaces can be written as $\{\mathbf{u}_1, \mathbf{u}_2\} \in \mathbb{R}^4$.

Let

$$\mathbf{W}^* = \begin{bmatrix} \mathbf{u}_2^\perp & \mathbf{u}_2^\perp & \mathbf{u}_1^\perp & \mathbf{u}_1^\perp \end{bmatrix}$$

where $\mathbf{u}_1^\perp, \mathbf{u}_2^\perp$ is some unit-length basis orthogonal to $\{\mathbf{U}_1, \mathbf{U}_2\}$ such that

$$\mathbf{u}_1^\top \mathbf{u}_1^\perp = 0, \quad \mathbf{u}_2^\top \mathbf{u}_2^\perp = 0$$

With that, the worst case would be

$$\tilde{\mathbf{X}}\mathbf{W}^* = \begin{bmatrix} \tilde{\mathbf{x}}_1^\top \\ \tilde{\mathbf{x}}_2^\top \\ \tilde{\mathbf{x}}_3^\top \\ \tilde{\mathbf{x}}_4^\top \end{bmatrix} \begin{bmatrix} \mathbf{u}_2^\perp & \mathbf{u}_2^\perp & \mathbf{u}_1^\perp & \mathbf{u}_1^\perp \end{bmatrix} = \begin{bmatrix} \Delta & \Delta & \delta & \delta \\ \Delta & \Delta & \delta & \delta \\ \delta & \delta & \Delta & \Delta \\ \delta & \delta & \Delta & \Delta \end{bmatrix}$$

Now, for the weight at $t$'th layer, we let $\mathbf{W}_K^t = \mathbf{W}_Q^t = \mathbf{W}^*$,

$$S^t = (\tilde{\mathbf{X}}\mathbf{W}_Q^t)(\tilde{\mathbf{X}}\mathbf{W}_K^t)^\top = 2 \begin{bmatrix} \Delta^2 + \delta^2 & \Delta^2 + \delta^2 & 2\Delta\delta & 2\Delta\delta \\ \Delta^2 + \delta^2 & \Delta^2 + \delta^2 & 2\Delta\delta & 2\Delta\delta \\ 2\Delta\delta & 2\Delta\delta & \Delta^2 + \delta^2 & \Delta^2 + \delta^2 \\ 2\Delta\delta & 2\Delta\delta & \Delta^2 + \delta^2 & \Delta^2 + \delta^2 \end{bmatrix}$$

Since the activation for the mid layer is just ReLU, we have the input for the next layer as:

$$\tilde{\mathbf{X}}^{t+1} := S^t \tilde{\mathbf{X}}^t = \begin{bmatrix} \alpha(\tilde{\mathbf{x}}_1^\top + \tilde{\mathbf{x}}_2^\top) + \beta(\tilde{\mathbf{x}}_3^\top + \tilde{\mathbf{x}}_4^\top) \\ \alpha(\tilde{\mathbf{x}}_1^\top + \tilde{\mathbf{x}}_2^\top) + \beta(\tilde{\mathbf{x}}_3^\top + \tilde{\mathbf{x}}_4^\top) \\ \alpha(\tilde{\mathbf{x}}_3^\top + \tilde{\mathbf{x}}_4^\top) + \beta(\tilde{\mathbf{x}}_1^\top + \tilde{\mathbf{x}}_2^\top) \\ \alpha(\tilde{\mathbf{x}}_3^\top + \tilde{\mathbf{x}}_4^\top) + \beta(\tilde{\mathbf{x}}_1^\top + \tilde{\mathbf{x}}_2^\top) \end{bmatrix}$$

where $\alpha := 2(\Delta^2 + \delta^2)$, $\beta := 4\Delta\delta$. Controlling the noise-level is key to achieving a high value for $\alpha$ and a low value for $\beta$. This is an important observation, as it indicates that $\tilde{\mathbf{X}}^{t+1}$ can effectively merge entries into the correct clusters when $\alpha$ is large. However, in order to fully establish the effectiveness of this approach, we must also demonstrate that the correlation within the same clusters is higher while the correlation between different clusters is lower. To this end, we assume that the weight of the next layer is also denoted by $\mathbf{W}_K = \mathbf{W}_Q = \mathbf{W}^*$.

When calculating similarity score of two entries from the different cluster:

$$\left(\alpha(\tilde{\mathbf{x}}_1^\top + \tilde{\mathbf{x}}_2^\top) + \beta(\tilde{\mathbf{x}}_3^\top + \tilde{\mathbf{x}}_4^\top)\right) \mathbf{W}_Q \mathbf{W}_K^\top \left(\alpha(\tilde{\mathbf{x}}_3 + \tilde{\mathbf{x}}_4) + \beta(\tilde{\mathbf{x}}_1 + \tilde{\mathbf{x}}_2)\right)$$
$$= \left(\alpha(\tilde{\mathbf{x}}_1^\top + \tilde{\mathbf{x}}_2^\top) + \beta(\tilde{\mathbf{x}}_3^\top + \tilde{\mathbf{x}}_4^\top)\right) 2 \left[\mathbf{u}_2^\perp (\mathbf{u}_2^\perp)^\top + \mathbf{u}_1^\perp (\mathbf{u}_1^\perp)^\top\right] \left(\alpha(\tilde{\mathbf{x}}_3 + \tilde{\mathbf{x}}_4) + \beta(\tilde{\mathbf{x}}_1 + \tilde{\mathbf{x}}_2)\right)$$

$$= \alpha\left(\tilde{\mathbf{x}}_1^\top + \tilde{\mathbf{x}}_2^\top\right) 2 \left[\mathbf{u}_2^\perp (\mathbf{u}_2^\perp)^\top\right] \alpha\left(\tilde{\mathbf{x}}_3 + \tilde{\mathbf{x}}_4\right) + \alpha\left(\tilde{\mathbf{x}}_1^\top + \tilde{\mathbf{x}}_2^\top\right) 2 \left[\mathbf{u}_1^\perp (\mathbf{u}_1^\perp)^\top\right] \alpha\left(\tilde{\mathbf{x}}_3 + \tilde{\mathbf{x}}_4\right)$$
$$+ \alpha\left(\tilde{\mathbf{x}}_1^\top + \tilde{\mathbf{x}}_2^\top\right) 2 \left[\mathbf{u}_1^\perp (\mathbf{u}_1^\perp)^\top\right] \beta(\tilde{\mathbf{x}}_1 + \tilde{\mathbf{x}}_2) + \alpha\left(\tilde{\mathbf{x}}_1^\top + \tilde{\mathbf{x}}_2^\top\right) 2 \left[\mathbf{u}_2^\perp (\mathbf{u}_2^\perp)^\top\right] \beta(\tilde{\mathbf{x}}_1 + \tilde{\mathbf{x}}_2)$$
$$+ \beta(\tilde{\mathbf{x}}_3^\top + \tilde{\mathbf{x}}_4^\top) 2 \left[\mathbf{u}_1^\perp (\mathbf{u}_1^\perp)^\top\right] \alpha\left(\tilde{\mathbf{x}}_3 + \tilde{\mathbf{x}}_4\right) + \beta(\tilde{\mathbf{x}}_3^\top + \tilde{\mathbf{x}}_4^\top) 2 \left[\mathbf{u}_2^\perp (\mathbf{u}_2^\perp)^\top\right] \alpha\left(\tilde{\mathbf{x}}_3 + \tilde{\mathbf{x}}_4\right)$$
$$+ \beta(\tilde{\mathbf{x}}_3^\top + \tilde{\mathbf{x}}_4^\top) 2 \left[\mathbf{u}_1^\perp (\mathbf{u}_1^\perp)^\top\right] \beta(\tilde{\mathbf{x}}_1 + \tilde{\mathbf{x}}_2) + \beta(\tilde{\mathbf{x}}_3^\top + \tilde{\mathbf{x}}_4^\top) 2 \left[\mathbf{u}_2^\perp (\mathbf{u}_2^\perp)^\top\right] \beta(\tilde{\mathbf{x}}_1 + \tilde{\mathbf{x}}_2)$$

$$\leq 16\alpha^2\delta + 16\alpha\beta\delta^2 + 16\alpha\beta + 16\beta^2\delta$$
$$= 16(\alpha^2 + \beta^2)\delta + 16\alpha\beta(1 + \delta^2)$$

When calculating similarity score of two entries from the same cluster:

$$\left(\alpha(\tilde{\mathbf{x}}_1^\top + \tilde{\mathbf{x}}_2^\top) + \beta(\tilde{\mathbf{x}}_3^\top + \tilde{\mathbf{x}}_4^\top)\right) \mathbf{W}_Q \mathbf{W}_K^\top \left(\alpha(\tilde{\mathbf{x}}_1 + \tilde{\mathbf{x}}_2) + \beta(\tilde{\mathbf{x}}_3 + \tilde{\mathbf{x}}_4)\right)$$
$$= \left(\alpha(\tilde{\mathbf{x}}_1^\top + \tilde{\mathbf{x}}_2^\top) + \beta(\tilde{\mathbf{x}}_3^\top + \tilde{\mathbf{x}}_4^\top)\right) 2 \left[\mathbf{u}_2^\perp (\mathbf{u}_2^\perp)^\top + \mathbf{u}_1^\perp (\mathbf{u}_1^\perp)^\top\right] \left(\alpha(\tilde{\mathbf{x}}_1 + \tilde{\mathbf{x}}_2) + \beta(\tilde{\mathbf{x}}_3 + \tilde{\mathbf{x}}_4)\right)$$

$$= \alpha\left(\tilde{\mathbf{x}}_1^\top + \tilde{\mathbf{x}}_2^\top\right) 2 \left[\mathbf{u}_2^\perp (\mathbf{u}_2^\perp)^\top\right] \alpha\left(\tilde{\mathbf{x}}_1 + \tilde{\mathbf{x}}_2\right) + \alpha\left(\tilde{\mathbf{x}}_1^\top + \tilde{\mathbf{x}}_2^\top\right) 2 \left[\mathbf{u}_1^\perp (\mathbf{u}_1^\perp)^\top\right] \alpha\left(\tilde{\mathbf{x}}_1 + \tilde{\mathbf{x}}_2\right)$$
$$+ \alpha\left(\tilde{\mathbf{x}}_1^\top + \tilde{\mathbf{x}}_2^\top\right) 2 \left[\mathbf{u}_1^\perp (\mathbf{u}_1^\perp)^\top\right] \beta(\tilde{\mathbf{x}}_3 + \tilde{\mathbf{x}}_4) + \alpha\left(\tilde{\mathbf{x}}_1^\top + \tilde{\mathbf{x}}_2^\top\right) 2 \left[\mathbf{u}_2^\perp (\mathbf{u}_2^\perp)^\top\right] \beta(\tilde{\mathbf{x}}_3 + \tilde{\mathbf{x}}_4)$$
$$+ \beta(\tilde{\mathbf{x}}_3^\top + \tilde{\mathbf{x}}_4^\top) 2 \left[\mathbf{u}_1^\perp (\mathbf{u}_1^\perp)^\top\right] \alpha\left(\tilde{\mathbf{x}}_1 + \tilde{\mathbf{x}}_2\right) + \beta(\tilde{\mathbf{x}}_3^\top + \tilde{\mathbf{x}}_4^\top) 2 \left[\mathbf{u}_2^\perp (\mathbf{u}_2^\perp)^\top\right] \alpha\left(\tilde{\mathbf{x}}_1 + \tilde{\mathbf{x}}_2\right)$$
$$+ \beta(\tilde{\mathbf{x}}_3^\top + \tilde{\mathbf{x}}_4^\top) 2 \left[\mathbf{u}_1^\perp (\mathbf{u}_1^\perp)^\top\right] \beta(\tilde{\mathbf{x}}_3 + \tilde{\mathbf{x}}_4) + \beta(\tilde{\mathbf{x}}_3^\top + \tilde{\mathbf{x}}_4^\top) 2 \left[\mathbf{u}_2^\perp (\mathbf{u}_2^\perp)^\top\right] \beta(\tilde{\mathbf{x}}_3 + \tilde{\mathbf{x}}_4)$$

$$\geq 8\alpha^2\Delta^2 + 8\beta^2\Delta^2$$
$$= 8(\alpha^2 + \beta^2)\Delta^2$$

Recall Definition 1 that the **sharpness** of the similarity score matrix $S$ is defined as the infimum of the ratio between the similarity scores of two points within the same cluster and the similarity scores of two points belonging to different clusters.

$$\mathcal{D}(S) := \inf_{i,j,k,h} \frac{S_{i,j}}{S_{k,h}}$$

where $x_i, x_j$ are from the same cluster and $x_k, x_h$ are from different clusters.

That means that

$$\mathcal{D}(S^t) = \frac{\alpha}{\beta}$$

$$\mathcal{D}(S^{t+1}) = \frac{(\alpha^2 + \beta^2)\Delta^2}{2(\alpha^2 + \beta^2)\delta + 2\alpha\beta(1 + \delta^2)}$$

Thus, the ratio of sharpness increased by layer $t$ is:

$$\frac{\mathcal{D}(S^{t+1})}{\mathcal{D}(S^t)} = \frac{(\alpha^2 + \beta^2)\Delta^2\beta}{2\alpha(\alpha^2 + \beta^2)\delta + 2\alpha^2\beta(1 + \delta^2)}$$

$\square$

