# OpenReview forum: "TransFusion: Contrastive Learning with Attention Layers"
_ICLR.cc/2024/Conference — ICLR 2024 Conference Desk Rejected Submission_

### Official Review · Reviewer_KE1w · 2023-10-29

**Soundness:** 1 poor
**Presentation:** 2 fair
**Contribution:** 2 fair
**Rating:** 3
**Confidence:** 3

**Summary:**

The paper proposes a contrastive learning framework named TransFusion. The core idea of the paper is fusing the features of different samples according to their classes. The proposed model consists of multiple customized self-attention blocks which compute the similarities between different samples and fuse them according to the similarities. In the final layer, the model is trained to minimize the KL-Divergence between the computed similarity matrix and the target similarity matrix based on classes of the sample.

**Strengths:**

(1) The paper is well-organized and easy to understand.

(2) The core idea is interesting and the motivation is clear.

**Weaknesses:**

One of the major drawbacks of this paper is lack of necessary experiments, the existing experimental results are too few to validate effectiveness of TransFusion. In this paper, all the numerical results can be used for comparison are only included in the Table 3, where the authors compare the classification accuracy between ResNet18 models trained with SupCon and TransFusion on CIFAR-10.

In the current experiment designs, the authors add an one-layer TransFusion to ResNet-18 pre-trained on ImageNet-1K and fine-tune the model on CIFAR-10 dataset. Then they extract intermediate output from the ResNet as embeddings, which are used as the input of SVM or KNN for final classification.

In my opinion, the authors should at least provide the following experimental results:

(1) Experimental results of models trained with TransFusion and more counterparts for comparison from scratch.

(2) Experimental results on different datasets. For example, SupCon provides results (training from scratch) on CIFAR-10, CIFAR-100 and ImageNet-1K. It also provides results (fine-tuning) on 12 natural image datasets.

(3) Experimental results of models trained with standard TransFusion. Since the authors have trained a 5-layer TransFusion model for the fashionMNIST dataset and 15-layer model for CIFAR10 dataset, I'm confusing why the authors only provide the t-sne visualization results without their classification accuracies. The existing classification results are obtained from the pre-trained ResNet-18 backbone with an one-layer TransFusion, which is different from the architecture of the TransFusion model shown in Figure 1.

(4) Experimental results of models trained with TransFusion without SVM and KNN. Only providing classification results using traditional machine learning methods is not enough.

(5) More ablation experiments. For example, the performance difference between TransFusion model using standard self-attention (between pixels) and proposed self-attention (between samples).

The other major drawback of TransFusion is the instability introduced by the self-attention mechanism between different samples. A TransFusion model consists of attention blocks which compute attention between different samples. It means that if we feed a batch of samples to a TransFusion model, the model may classify a sample into different classes if the other samples change. I'm worry about the effect of "batch" on performance of TransFusion. Such instability will make the design unpractical.

**Questions:**

I'm wondering about the effect of "batch" on performance of TransFusion.

(1) How the number of classes affects the performance of TransFusion model? When the number of classes gets larger, there will be much fewer samples in a batch which belong to the same class. Will the self-attention mechanism between samples still work under such condition?

(2) How the batch size affects the inference speed of TransFusion model?

(3) How the training batch size affects the performance of TransFusion model?

(4) How the testing batch size affects the performance of TransFusion model? Speficially, if we set batch size to 1, will the performance of TransFusion model sharply decrease?

(5) I'm also wondering the mean and variance of classification accuracies of TransFusion model with different random permutations of samples in a dataset.

---

### Official Review · Reviewer_Ak7r · 2023-10-30

**Soundness:** 1 poor
**Presentation:** 1 poor
**Contribution:** 2 fair
**Rating:** 3
**Confidence:** 4

**Summary:**

The paper presents TransFusion, for training attention-based neural networks to extract features for classification tasks. It adopts a strategy of training with affinity matrices, effectively capturing the resemblances among samples within the same class. The authors present theoretical proof for their design.

**Strengths:**

- Well written and easy to understand the idea.

- Good theoretical proofs.

**Weaknesses:**

Totally, I think this is not a paper but a report.

- The novelty issue: the proposed transformer layer is nothing but a standard self-attention with more residual links.

- The experiment part is not convincing. Only one result on CIFAR-10 is not enough.
What are about the ImageNet results? Where are the ablation studies?

- Better re-draw the Figure-1 and make it easy to understand. Better remove the Tab.1.

- Better re-write the entire paper since most words are not academic.

**Questions:**

See the weakness.

---

### Official Review · Reviewer_VzcH · 2023-11-05

**Soundness:** 3 good
**Presentation:** 3 good
**Contribution:** 2 fair
**Rating:** 3
**Confidence:** 4

**Summary:**

This paper introduces a novel framework for training attention-based neural networks to extract meaningful features for downstream classification tasks. The authors leverage the fusion-like behavior of the self-attention mechanism in TransFusion, where samples from the same cluster have higher attention scores and gradually converge and propose training with affinity matrices to capture resemblances among samples within the same class. This paper provides theoretical insights into the behavior of each layer and demonstrates TransFusion's effectiveness in fusing data points within the same cluster while managing noise levels. Experimental results indicate that TransFusion successfully extracts features that isolate clusters from data, leading to improved classification accuracy in downstream tasks.

**Strengths:**

1). This paper introduces a fusion model based on the self-attention mechanism, which effectively captures features among samples within the same class. The weighted-sum operation fuses samples from the same cluster, which leads to denser and distinct clusters.

2). This paper analyzes the behavior of each layer of TransFusion, which demonstrates how they contribute to the fusion process and manage noise levels. The TransFusion proposed in this paper establishes an upper limit on the required augmentation for successful fusion and highlights the influence of class structure on fusion effectiveness.

3). Experimental results show that TransFusion successfully extracts features that isolate clusters from complex real-world data, resulting in improved classification accuracy in downstream tasks.

**Weaknesses:**

1). This paper should provide more ablation experiments to demonstrate the feasibility and accuracy of each theory of the method provided and its effectiveness under different settings.

2). The presentation of this paper could be improved. It is relatively difficult to read for the interpretation part of the method, and it is difficult for readers to efficiently obtain useful information from the article. The author should use as clear a figure and process expression as possible to make the specific content statement of the method clearer and provide more implementation details to improve the reproducibility of the paper.

3). The author should show the effect of this method on more different types of datasets and more difficult datasets to show the robustness of the method in different scenarios and different problems. In my opinion, the experiments and comparisons provided by this article are insufficient. In particular, the effect of contrastive learning is usually more effective on large-scale datasets. However, all the experiments conducted in this paper are with very small datasets and small network architectures. In this case, the drawed conclusions might not be generalizable.

**Questions:**

See the weakness above.

---

### Official Review · Reviewer_WPJh · 2023-11-07

**Soundness:** 3 good
**Presentation:** 3 good
**Contribution:** 2 fair
**Rating:** 5
**Confidence:** 4

**Summary:**

This paper proposes *TransFusion* to enhance feature extraction in the transformer networks, which is a fusion-like self-attention module to model the inter-sample relation. The goal of TransFusion is that samples belonging to the same class should be closer to each other in distance compared to samples from different classes. Experiments conducted on FasionMNIST and CIFAR10 datasets demonstrate the effectiveness of the proposed method.

**Strengths:**

- This paper validates the importance to model inter-sample relation during model training.
- Theoretical analysis is provided.

**Weaknesses:**

- It is not the first time to use self-attention module to model inter-sample relation, such as BatchFormer [1][2]. However, this paper didnot provide an in-depth discussion about the difference of the proposed method and the existing works.

- The existing works [1][2] demonstrate the effectiveness of the application of self-attention module in modeling inter-sample relation on many benchmarks, including self-supervised representation learning, domain generalization, long-tail recognizion, object detection, etc. However, this paper only conducts experiments on two very simple image classification tasks, FasionMNIST and CIFAR10, which I think is not enough.

[1] BatchFormer: Learning to Explore Sample Relationships for Robust Representation Learning. CVPR 2022

[2] BatchFormerV2: Exploring Sample Relationships for Dense Representation Learning. Arxiv 2022

**Questions:**

See the weaknesses.